# SG-Adapter: Enhancing Text-to-Image Generation with Scene Graph Guidance

## Abstract

Recent advancements in text-to-image generation have been propelled by the development of diffusion models and multi-modality learning. However, since text is typically represented sequentially in these models, it often falls short in providing accurate contextualization and structural control. So the generated images do not consistently align with human expectations, especially in complex scenarios involving multiple objects and relationships. In this paper, we introduce the **Scene Graph Adapter** (SG-Adapter), leveraging the structured representation of scene graphs to rectify inaccuracies in the original text embeddings. The SG-Adapter's explicit, non-fully connected graph representation significantly improves upon the causal connections commonly used in transformer-based text models. In causal connections, each token can attend to all previous tokens, which may result in attribute leakage. On the other hand, we also curated a highly clean, multi-relational scene graph-image paired dataset MultiRels to address the challenges posed by low-quality annotated datasets like Visual Genome (Krishna et al., 2017). Furthermore, we design three metrics derived from GPT-4V(Achiam et al., 2023) to effectively and thoroughly measure the correspondence between images and scene graphs. Both qualitative and quantitative results validate the efficacy of our approach in controlling the correspondence in multiple relationships.

## 1 Introduction

Image generation has made great progress thanks to the success of a series of text-to-image diffusion models (Rombach et al., 2022; Saharia et al., 2022; Ramesh et al., 2022; Ho et al., 2020; Nichol et al., 2021; Dhariwal & Nichol, 2021; Song et al., 2020). Due to the huge amount of text-image paired training data (Schuhmann et al., 2022; Lin et al., 2014) and the numerous model parameters, these text-conditioned models show fantastic image quality and attract great attention from researchers as well as industries.

However, the text encoder used by these models are suboptimal for generating coherent images. They often face challenges in contextualizing text tokens, which requires capturing both their intrinsic meanings and their contextual understanding from surrounding tokens. The most commonly used text encoder, the CLIP model (Radford et al., 2021), applies causal attention mechanisms to allow tokens to gather information from preceding ones. However, this sequential processing can lead to the "leakage issue", where relations or attributes incorrectly influence unrelated elements in the image. For example, in "A man playing the guitar back to back with a woman", the relation "playing guitar" might be mistakenly applied to the woman due to the linear text sequence (Fig. 1). Such cases, where the encoder fails to recognize distinct entity boundaries, underscore the need for more advanced text embedding methods capable of accurately depicting separate entities and their respective relations in complex scenes.

Scene graphs, with their structured representation, effectively avoid the contextualization issues inherent in sequential text tokens. Scene graphs depict images as networks of entities (nodes) and their relationships (edges), ensuring clear, non-linear associations. Relations are directly linked to specific nodes, preventing the ambiguity and misinterpretation common in text-based processing. This distinct structure allows for precise and unambiguous representation of complex scenes, enhancing the accuracy of image generation. This capability positions them as a natural control signal for image generation.

Figure 1: **Overcoming Contextualization Limits in Image Generation with Scene Graph**. The left section highlights the limitations of text embeddings in sequential text processing, showcasing how relations like "playing guitar" may erroneously apply to the "woman". The right section illustrates the improvements of using a Scene Graph, which provides structured clarity, enabling precise relation.

However, pure scene graph-to-image generation (Yang et al., 2022a; Johnson et al., 2018; Farshad et al., 2023; Li et al., 2019) lags behind text-to-image generation in terms of image quality and is far from practical use. This is because the scene graph-image pair data are considerably smaller than text-image pair data. For instance, Visual Genome(Krishna et al., 2017) includes 108,077 pairs with limited-quality scene graph labels. In stark contrast, the typical text-image dataset LAION-5B (Tang et al., 2020) contains a massive 5.85 billion CLIP-filtered image-text pairs.

Acknowledging the challenges in text-to-image generation and scene graph datasets, our research aims to leverage the limited scene graph annotations to enhance the control and accuracy of text-to-image generation. Recent studies on adapters (Li et al., 2023a; Zhang et al., 2023; Mou et al., 2023), in pre-trained text-to-image models offer promising directions. These adapters introduce new control mechanisms that retain the model's image generation quality.

In this study, we re-examine the text-to-image generation process, pinpointing issues of incorrect contextualization arising during text encoder computation. This problem is traced back to the causal attention mask, which is used in the pre-trained CLIP text model (Radford et al., 2021). It fails to account for the structural semantics inherent in captions. However, directly replacing this causal attention mask during inference is ineffective through our observations, as presented in Appendix.A.1. Consequently, we propose the Scene Graph Adapter (SG-Adapter), strategically plugged in after the CLIP text encoder, to tackle this issue. This adapter utilizes scene graph knowledge to refine text embeddings. Critically, to ensure precise contextualization, we design our adapter based on a transformer architecture and incorporate a novel triplet-token attention mask called Scene Graph(SG) Mask, fostering accurate and contextually relevant text-to-image generation.

On the other hand, scene graph annotations often face challenges such as limited rationality, language biases, or reporting biases, resulting in visual relationships that are often simplistic and less informative (Tang et al., 2020). These less reliable annotations can complicate experimental validation. To address this, we developed a multi-relational dataset, MultiRels, with highly curated annotations that better capture complex semantic structures. Using this high-quality dataset, we clearly demonstrate that our SG-Adapter excels in generating relations with precise correspondences, showcasing its strong ability to capture complex relational structures. Moreover, considering traditional evaluation metrics like FID and CLIP-Score cannot perceive the abstract Relation Correspondence concept, we propose three evaluation metrics derived from the advanced GPT-4V(Achiam et al., 2023) to evaluate the correspondences between images and scene graphs. In summary, our contributions are as follows:

- We propose our SG-Adapter to correct the incorrect contextualization in text embeddings that results in "relation leakage". Our adapter effectively addresses this issue and enhances the structural semantics generation capabilities of current text-to-image models.
- We create a more comprehensive dataset with high-quality annotations called MultiRels that could be adopted to better demonstrate the generation models' ability to handle complex structural semantics. Besides, for effective and fair comparison in terms of relation accuracy, we contribute three metrics derived from GPT-4V (Achiam et al., 2023).
- Both qualitative and quantitative results illustrate that our SG-Adapter outperforms state-of-the-art text-to-image and SG-to-image methods in terms of relation generation and control.

## 2    RELATED WORK

**Text-to-Image Generation.** The evolution of text-to-image diffusion models (Rombach et al., 2022; Saharia et al., 2022; Ramesh et al., 2022; Ho et al., 2020; Nichol et al., 2021; Dhariwal & Nichol, 2021; Song et al., 2020) has marked a significant advancement in the generation of high-quality images. Among these, Stable Diffusion (SD) (Rombach et al., 2022) is particularly notable, utilizing a pre-trained autoencoder alongside a diffusion model to effectively transform textual prompts into precise and detailed visual representations. Despite their impressive capabilities, these models still face challenges, particularly in terms of the leakage issue, where relations or attributes inadvertently influence unrelated elements within an image (Feng et al., 2022; Li et al., 2023b). While (Li et al., 2023b) addresses this through a novel Jensen-Shannon divergence-based binding loss, it does not fully acknowledge the issue as stemming from the improper contextualization of text embedding. In a similar vein, (Feng et al., 2022) employs linguistic structures such as constituency trees or scene graphs to guide the diffusion process. However, this method does not entirely address the root cause of these issues but constructs a set of text embeddings and only partially fixes the wrong contextualization, which is linked to the causal attention mechanism used in the CLIP text encoder.

**Scene Graph-to-Image Generation.** In the realm of scene graph to image generation, some works often rely on scene layouts as an intermediate representation (Johnson et al., 2018; Farshad et al., 2023) in a two-stage pipeline. They first create scene layouts from scene graphs and then generate images based on the layouts. These layouts, serving as image-like representations of scene graphs, can often be suboptimal due to manual crafting and potential misalignment with actual scene graphs. In this paper, we clarify that **Layout-to-Image** (Zheng et al., 2023; Li et al., 2023a; Chen et al., 2023; Couairon et al., 2023) **and Scene Graph-to-Image are separate different topics**, as demonstrated in Fig.5. Layout-to-Image aims to generate objects in user-specified areas while having limited ability to represent complicated Relations. This limitation has led to the development of alternative approaches, such as SGDiff (Yang et al., 2022b), which optimizes scene graph embeddings for better alignment with images, and other works (Wu et al., 2023) that introduce "knowledge consensus" as a means to disentangle complex semantics between knowledge graphs and images. Despite the advances in this field, the dearth of large-scale and high-quality data continues to impede its widespread practical application.

**Single Relation Learning.** Though Text-to-Image generation models have made great progress, they might fall short when generating complicated single relationships. RRNet (Wu et al., 2024) learns a relationship via a heterogeneous GCN while Reversion (Huang et al., 2023) reverses the target relation from example images by optimizing the relation prompt. Instead of addressing the issue of single relations, our study focuses on generating multiple relations with accurate correspondences.

**Adapter.** The integration of adapters into existing models has emerged as a noteworthy innovation, exemplified by methods such as ControlNet (Zhang et al., 2023) and GLIGEN (Li et al., 2023a). These techniques have enabled models like SD to enhance user control without requiring extensive retraining. For example, ControlNet (Zhang et al., 2023) operates as a plug-in module, extracting and integrating residual features from each image condition into SD's U-Net for enhanced control. GLIGEN (Li et al., 2023a), on the other hand, incorporates location control by integrating a new self-attention module into the U-Net.

Our work critically examines the issue of false contextualization in SD and develops a scene graph-only trained adapter to enhance the semantic structure controllability of SD, addressing a gap in the current landscape of text-to-image generation.

## 3    PROPOSED METHOD

In this work, we aim to rectify the incorrect contextualization in text embedding caused by the prevalent causal attention masks in language models, *i.e.*, CLIP text encoder $E_T(\cdot)$.

### 3.1    DISCUSSIONS OF CAUSAL ATTENTION

The causal attention mask in the context of transformer models ensures that the prediction for a specific token can only consider previously generated tokens. For a caption $c$ of $N$ tokens, the causal attention mask $\mathbf{M}$ is an $N \times N$ matrix where each entry $M_{ij}^{\text{causal}}$ is defined as:

$$\mathbf{M}_{ij}^{\text{causal}} = \begin{cases} 0, & \text{if } j \leq i, \\ -\infty, & \text{if } j > i. \end{cases} \tag{1}$$

This matrix is utilized in the scaled dot-product attention mechanism within the transformer model. The attention scores with the mask are computed as follows:

$$\text{Attention}(\mathbf{Q}, \mathbf{K}, \mathbf{V}, \mathbf{M}) = \text{softmax}\left(\frac{\mathbf{Q}\mathbf{K}^T}{\sqrt{d_k}} + \mathbf{M}^{\text{causal}}\right)\mathbf{V}, \tag{2}$$

where $\mathbf{Q}$, $\mathbf{K}$, and $\mathbf{V}$ denote the query, key, and value matrices respectively, and $d_k$ represent the dimension of the key vectors that are used for scaling. The softmax function is applied subsequent to the addition of the causal attention mask $\mathbf{M}^{\text{causal}}$. This ensures that for each position $i$ in the sequence, the attention is only on positions $j \leq i$, effectively masking out the future tokens by assigning them a value that approximates zero probability in the softmax due to the $-\infty$ entries in the mask.

However, the causal attention mask may fail to maintain the integrity of subject-relation-object bindings within captions, leading to incorrect contextual associations between entities and relations. For instance in Fig. 2, the caption "a man holds a cake and a woman holds an apple" contains two distinct subject-relation-object bindings. However, traditional causal attention will not mask out the former token "holds a cake" when computing the embedding of "a woman" and may incorrectly leak the semantics of "holds a cake" to "a woman", disrupting the semantic structure of the caption.

To enable precise interactions among tokens, a thorough understanding of the semantic structure encapsulated in the caption is crucial. This structure is aptly represented by a scene graph, constituted of a sequence of triplets formatted as ⟨subject, relation, object⟩. The extraction of these $K$ triplets from the caption can be achieved either through an NLP parser (Feng et al., 2022) or GPT-4 (Achiam et al., 2023). Each triplet $\mathcal{T}_k$ is indexed by $k$, and a token-to-triplet mapping function $\tau(\cdot)$ is defined, which maps each token $i$ to its respective triplet index. The prompt used in GPT-4 is provided in Appendix. A.5.

Our analysis stipulates that interaction between any two tokens $i$ and $j$ is permissible solely when they are constituents of an identical triplet, as determined by the condition $\tau(i) = \tau(j)$. In light of this, we propose refining the causal attention mask such that it facilitates the alignment of each token with its associated triplet. This refinement process is intended to bolster the model's ability to discern and maintain contextual relationships. The Triplet-Triplet attention mask $\mathbf{M}^\tau$, adjusted for triplet alignment, is formally given by:

$$\mathbf{M}_{ij}^\tau = \begin{cases} 0, & \text{if } \tau(i) = \tau(j), \\ -\infty, & \text{otherwise.} \end{cases} \tag{3}$$

With the refined attention mask, the most straightforward approach would be to replace the CLIP attention mask during inference. However, this can cause out-of-distribution issues, as shown in the Appendix. A.1. Additionally, training a CLIP model from scratch using a scene graph attention mask requires substantial resources.

## 3.2 Scene Graph Guided Generation

**Scene Graph Representation.** To address these challenges, we devised an alternative approach that augments the text embeddings produced by the CLIP model with the scene graph information. The text embedding output by the CLIP model is denoted as $\mathbf{w} = E_T(c)$, where $\mathbf{w} \in \mathbb{R}^{N \times D}$ and $D$ signifies the dimensionality of the embedding vectors by CLIP Text Encoder $E_T$. Here, $\mathbf{w}_i \in \mathbb{R}^D$ represents the embedding of the $i^{th}$ token.

To encapsulate the scene graph information, we construct a unified embedding for each semantic triplet. For a given triplet $\mathcal{T}_k = \langle s_k, r_k, o_k \rangle$, where $s_k$, $r_k$, and $o_k$ denote the subject, relation, and object of the triplet respectively, we apply $E_T$ to each component to obtain their embeddings. These embeddings are then concatenated to form a composite triplet embedding as follows:

$$\mathbf{e}_k = l(\text{concat}(E_T(s_k), E_T(r_k), E_T(o_k))), \tag{4}$$

where $l(\cdot)$ is a projection function that maps its input to the dimensionality $D$. The final scene graph embedding $\mathbf{e} \in \mathbb{R}^{K \times D}$ is the concatenation of all the triplet embeddings. This embedding is then utilized to refine the embeddings of corresponding tokens.

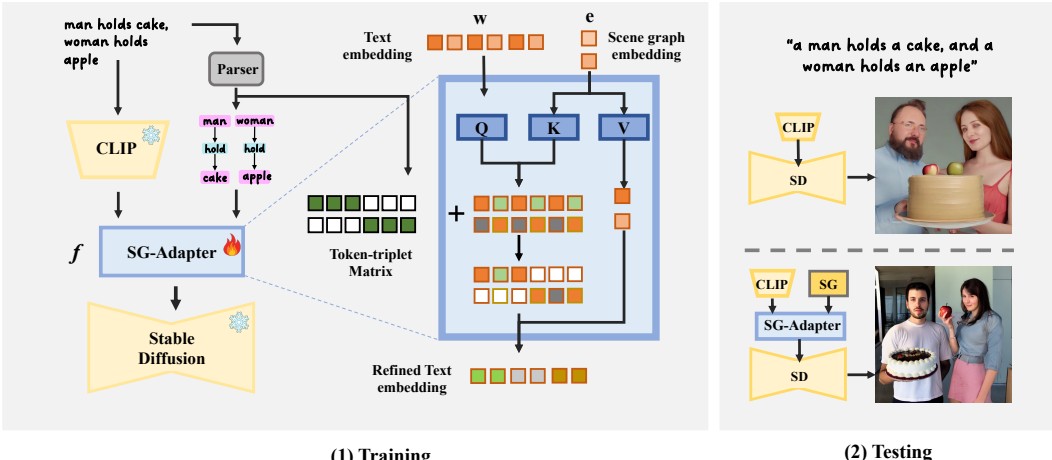

**(1) Training**  **(2) Testing**

Figure 2: **Framework for SG-Adapter in Stable Diffusion.** The **Parser** (could be either an NLP tool (Feng et al., 2022) or GPT-4 (Achiam et al., 2023)) extracts linguistic structures from text inputs. **Scene graph embeddings** are computed as per equation 4. The **token-triplet matrix**, generated by the function $\tau$, guides the refinement of each token and its associated triplet. During testing, when integrated with our SG-Adapter, Stable Diffusion more accurately captures the intended semantic structure in the generated images.

**Scene Graph Adapter.** To implement the refinement, an adapter $f(\cdot)$ is designed. Conceptualized as a transformer module, our adapter predominantly integrates a cross-attention layer. This layer facilitates relation-text attention by leveraging the scene graph embedding $\mathbf{e}$ to calculate the keys $\mathbf{K}$ and values $\mathbf{V}$, and the text embedding $\mathbf{w}$ to calculate the queries $\mathbf{Q}$. Such interaction is pivotal in updating the text embeddings with relation-specific nuances. The model $f$ is mathematically represented as:

$$\begin{aligned} \mathbf{w}' &= f(\mathbf{w}, \mathbf{e}, \mathbf{M^{sg}}) \\ &= \text{Attention}(\mathbf{Q}, \mathbf{K}, \mathbf{V}, \mathbf{M^{sg}}), \end{aligned} \tag{5}$$

where $\mathbf{Q} = l_Q(\mathbf{w})$, $\mathbf{K} = l_K(\mathbf{e})$, and $\mathbf{V} = l_V(\mathbf{e})$ are derived using respective projection layers in the attention framework. The term $\mathbf{w}'$ represents the improved text embeddings resulting from the application of the cross-attention mechanism.

To ensure that each token embedding $\mathbf{w}_i$ attends to the appropriate triplet embedding $\mathbf{e}_{\tau(i)}$, we introduce the Token-Triplet attention mask $\mathbf{M^{sg}} \in \mathbb{R}^{N \times K}$. This mask is designed to allow each token embedding to attend exclusively to its corresponding triplet embedding. The mask is formalized as follows:

$$\mathbf{M}^{\mathbf{sg}}_{ik} = \begin{cases} 0, & \text{if } \tau(i) = k \\ -\infty, & \text{otherwise} \end{cases} \tag{6}$$

and implementing this mask within the cross-attention layer of our model permits $\mathbf{w}_i$ to be selectively refined based on the contextual relevance of $\mathcal{T}_{\tau(i)}$. Such a targeted approach ensures precise contextualization within the generated scene graph.

To train our model, we leverage the pre-trained diffusion model framework. Given an image $x$, its corresponding text embedding $\mathbf{w}$, and the extracted scene graph embedding $\mathbf{e}$, our model undergoes a training phase analogous to that of the diffusion model. The objective is to minimize the discrepancy between the predicted and actual noise variables. The training loss function is defined as:

$$\mathcal{L}_t = \mathbb{E}_{\mathbf{x}, t, \epsilon} \left[ \| \epsilon_t - \epsilon_\theta(\mathbf{x_t}, t, f(\mathbf{w}, \mathbf{e}, \mathbf{M^{sg}})) \|_2^2 \right], \tag{7}$$

where $\epsilon_t$ represents the noise variable at time step $t$, and $\epsilon_\theta$ denotes the noise prediction from the pre-trained diffusion model parameterized by $\theta$. The function $f(\mathbf{w}, \mathbf{e}, \mathbf{M^{sg}})$ encapsulates the adapter's output, which refines the text embedding with relation-specific context derived from the scene graph embedding. Our training algorithm iteratively adjusts the parameters of $f(\cdot)$ to reduce $\mathcal{L}_t$, thereby improving the fidelity of generated images to the input descriptions and scene graph structure.

# 4 EXPERIMENTS

## 4.1 DATASET CONFIGURATION

As we mentioned above, a scene graph dataset with clean and precise relational annotations is essential as well as effective for learning relation representation. To train a model that can encode both single and multiple relations accurately, for the first time, we present a multiple relations scene graph-image paired dataset with highly precise labels called **MultiRels**. The dataset with size 309 is composed of two parts as follows:

- ReVersion (Huang et al., 2023): This part consists of 99 images with only a single clear relation. There are 10 representative relations in Reversion and most of them are "difficult" relations that current text-to-image models can not generate well, e.g., *shake hands with*, *sit back to back*, *is painted on*. Reversion is mainly used to learn new single relations.
- Multiple Relations: For multi-relations learning, in this part, we manually collect images with 1-4 (mostly more than 1) salient relations and label them with accurate scene graphs. Most of the relations are "simple" relations (e.g., *holding*, *drinking*, *stands on*, *sits on*) that current text-to-image models generate well individually but fail when existing multiple objects and multiple relations. There are also small parts of relations that are "difficult" like *is above*, *is under*. Multiple Relations is towards to train a model that could generate multiple relations with correct correspondences.

Besides the scene graph, we additionally provide a token-triplet matrix for each image.

**Test Scenarios.** To fairly validate the effectiveness and generalization performance of our method, we design 20 different testing scenarios that regroup 2-3 randomly selected relations in MultiRels. For more details about the MultiRels, please refer to our Appendix.

## 4.2 BASELINE METHODS

In our experiments, we evaluate the performance of SG-Adapter against several alternative methods, each with a distinct approach to text-to-image generation:

- **Fine-tuning (FT):** No adapters are used. We directly fine-tune components of Stable Diffusion, CLIP text encoder according to our analysis in Sec.1.
- **LoRA Adapter:** Utilizes the LoRA (*Low-Rank Adaptation*) adapter (Hu et al., 2021). The LoRA adapter does not integrate direct scene graph information in the adaptation process.
- **Gated Self-Attention Adapter:** Employs the adapter proposed by GLIGEN (Li et al., 2023a) which introduces scene graph tokens into the model. For generating scene graph tokens, we use a specialized scene graph encoder that has been pre-trained for this specific task (Yang et al., 2022b).

Note these baseline methods also assess the impact of Scene Graph Embedding from two critical angles:

- **Inclusion of SG Embed**: We explore the necessity of SG Embed by contrasting the performance of our SG-Adapter with the LoRA Adapter, which operates without SG Embed.
- **Placement of SG Embed**: We assess the effect of embedding placement by comparing our model to the GLIGEN Adapter, which integrates SG Embed within a U-Net architecture.

## 4.3 QUALITATIVE EVALUATION

We begin with a qualitative assessment, showcasing synthetic images from each method. SG-Adapter's ability to accurately depict complex relational structures and entities is highlighted, with Fig. 3 illustrating its superiority in maintaining correct relation correspondence in generated images. More qualitative results are provided in our Appendix.

## 4.4 QUANTITATIVE EVALUATION

**Automatic Metrics.** Since relation correspondence in images is complicated and abstract, existing metrics like FID can not perceive such conception. To fairly and effectively evaluate the ability to generate accurate relations of each method, we contribute three metrics derived from the advanced GPT-4V(Achiam et al., 2023): Scene Graph(SG)-IoU, Relation-IoU, and Entity-IoU. These metrics

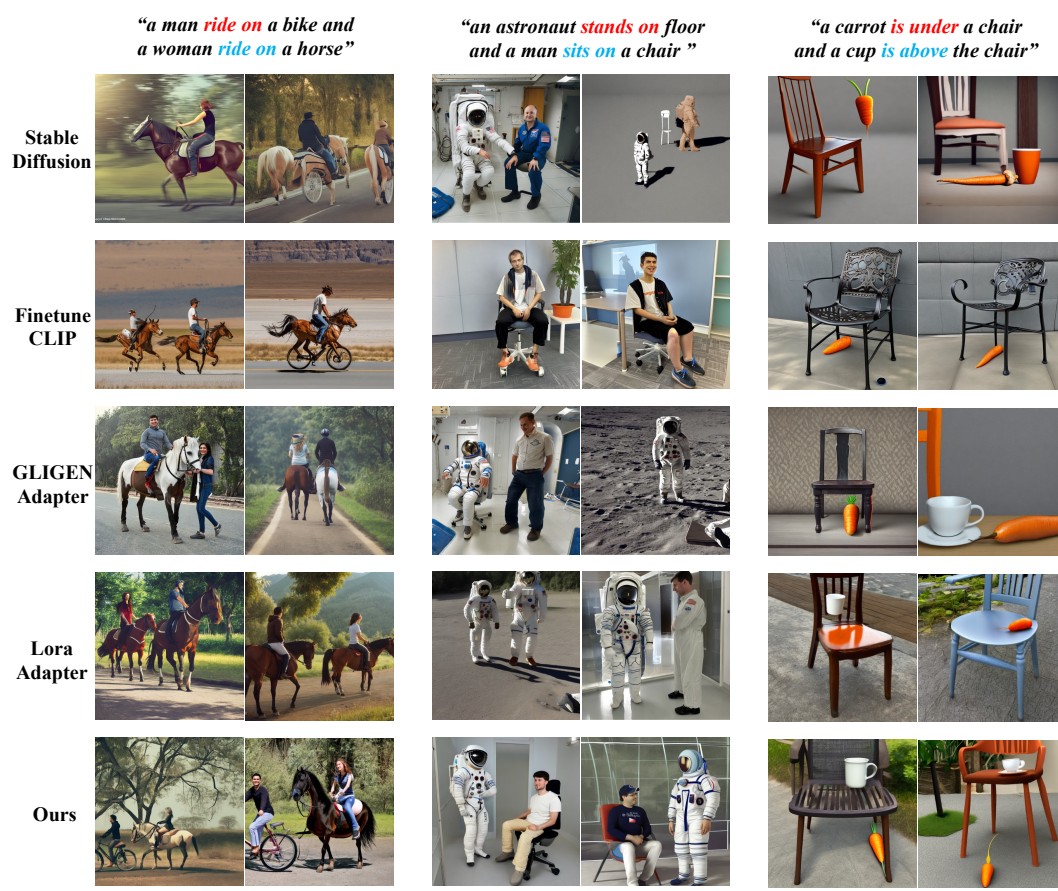

Figure 3: **Qualitative Comparisons** with Adaptation Methods. In addition to precisely generating each individual relation in the text prompt, our SG-Adapter successfully creates all multiple relations together in correct correspondence.

Table 1: **Quantitative Evaluation** of each method in terms of Automatic Relational Metrics, Human Evaluations, and Image Quality.

| Method | Automatic Metrics | | | Human Evaluations | | FID ↓ |
|---|---|---|---|---|---|---|
| | SG-IoU ↑ | Entity-IoU ↑ | Relation-IoU ↑ | Relation-Accuracy ↑ | Entity-Accuracy ↑ | |
| Stable Diffusion (Rombach et al., 2022) | 0.157 | 0.673 | 0.526 | 5.38% | 5.48% | **25.0** |
| Finetune CLIP | 0.198 | 0.499 | 0.635 | 5.38% | 6.78% | 58.2 |
| GLIGEN Adapter (Li et al., 2023a) | 0.141 | 0.689 | 0.546 | 5.72% | 5.58% | 27.4 |
| LoRA Adapter (Hu et al., 2021) | 0.145 | 0.653 | 0.540 | 5.96% | 5.05% | 27.5 |
| **SG-Adapter** | **0.623** | **0.812** | **0.753** | **77.6%** | **77.1%** | 26.2 |

are computed as follows: Given a generated image $I$, we obtain the scene graph represented as a list of triplets: $\hat{\mathcal{T}} = \{\hat{\mathcal{T}}_1, \hat{\mathcal{T}}_2, \ldots, \hat{\mathcal{T}}_n = \langle \hat{s}_n, \hat{r}_n, \hat{o}_n \rangle\}$ and the entity list $\hat{\xi}$ using GPT-4V, denoted as $\hat{\mathcal{T}}, \hat{\xi} = \text{GPT}(I)$. The extraction prompt is provided in the appendix, see A.5. From $\hat{\mathcal{T}}$, we can also derive the relation list $\hat{\mathbf{r}}$. Then, given the input scene graph $\mathcal{T}$ and entity list $\xi$ derived from $\mathcal{T}$, the three metrics are computed as follows:

$$\text{SG-IoU} = \text{IoU}(\mathcal{T}, \hat{\mathcal{T}}), \quad \text{Entity-IoU} = \text{IoU}(\xi, \hat{\xi}), \quad \text{Relation-IoU} = \text{IoU}(\mathbf{r}, \hat{\mathbf{r}})$$

SG-IoU indicates whether each relationship is well generated according to input correspondence and the other two metrics show whether each object and relation are generated.

Table 2: Ablation study on SG Mask.

| Method | SG-IoU ↑ | Entity-IoU ↑ | Relation-IoU ↑ | FID ↓ |
|---|---|---|---|---|
| **w/o SG Mask** | 0.316 | 0.742 | 0.668 | 26.7 |
| **SG-Adapter** | 0.623 | 0.812 | 0.753 | 26.1 |

**Human Evalution.** We conduct a user study engaging **104** participants in evaluating **20** test scenarios, each comprising textual descriptions and corresponding sets of images from the evaluated methods. We collect the percentage of user preference for each method in terms of the following criteria:

*Entity Accuracy*: Evaluating the precision in representing all textual entities within the images.

*Relation Accuracy*: Assessing the accuracy of depicted relationships between entities, ensuring alignment with the textual descriptions.

**Image Quality.** We compute the FID(Heusel et al., 2018) which quantifies the discrepancy between the distribution of generated images and 5000 validation images from the MS-COCO-Stuff (Caesar et al., 2018).

**Quantitative Analysis.**

- *Relation Correctness.* As demonstrated in Tab. 1, SG-Adapter consistently outperforms the baseline methods on both Automatic Metrics and Human Evaluations, significantly on SG-IoU and Relation Accuracy, showing a strong ability to generate relations with precise correspondences. Besides, high Entity-IoU and Relation-IoU while low SG-IoU indicate that though all the baseline methods successfully generated the required entity and relation, they can not distribute them to each other with correct correspondence.
- Image Quality Maintenance. Finetuning a pre-trained T2I model on a relatively small dataset will degrade the FID inevitably(Wang et al., 2023; Ruiz et al., 2023). As presented in Tab.1, our SG-Adapter maintains the image quality best among all the alternatives.

### 4.5 ABLATION STUDY

To elucidate the contributions of SG-Adapter's distinct components, we performed a comparative analysis by systematically ablating specific features. Through our experiments, we have demonstrated that the inclusion of scene graph embeddings is essential and that integrating them into the text encoder is more effective. We conduct an ablation study on the internal elements of our method:

**Token-to-Triplet Causal Mask (SG Mask)**: We assessed the significance of the causal mask designed to map tokens to triplets, evaluating its role in refining the image generation process.

As shown in Tab. 2, our method performs better than the one without SG mask on all evaluation metrics, indicating the effectiveness of the proposed SG mask.

### 4.6 SG-TO-IMAGE GENERATION EVALUATION

We further benchmark our method against state-of-the-art SG-to-Image generation approaches including SG2IM (Johnson et al., 2018), PasteGAN (Li et al., 2019), SceneGenie (Wu et al., 2023), SGDiff (Yang et al., 2022b), R3CD(AAAI 2024) (Liu & Liu, 2024), and ISGC(Arxiv 2024) (Mishra & Subramanyam, 2024) both quantitatively and qualitatively. (Note that SceneGenie, R3CD and ISGC do not release the code, we just report the quantitative result from their paper.) We train our SG-Adapter on the commonly used Scene Graph dataset (Plummer et al., 2015) to demonstrate that our method is extendable to large-scale datasets. For Flickr30k (Plummer et al., 2015), we employed NLP processing techniques (Feng et al., 2022) to extract scene graph information.

For qualitative comparison, we present generated images of each method in Fig. 4. Benefiting from our learned adapter strategy, we harness the high-resolution and high-fidelity generative capabilities of pre-trained text-to-image models, thereby achieving superior image quality in our results. Moreover, our method adeptly renders complex relational structures in images, signifying that our adapter effectively leverages scene graph information to accurately reflect intricate relationships within the visual output.

Table 3: Comparison with SG2I Generation Methods.

| Method | SG2IM | PasteGAN | SGDiff | SceneGenie | R3CD(AAAI24) | ISGC(24) | SG-Adapter |
|---|---|---|---|---|---|---|---|
| **FID** ↓ | 99.1 | 79.1 | 36.2 | 62.4 | 32.9 | 38.1 | **25.1** |
| **Inception Score** ↑ | 8.20 | 12.3 | 17.8 | 21.5 | 19.5 | 30.2 | **57.8** |
| **SG-IoU** ↑ | 0.085 | 0.091 | 0.122 | - | - | - | **0.413** |
| **Entity-IoU** ↑ | 0.297 | 0.382 | 0.436 | - | - | - | **0.729** |
| **Relation-IoU** ↑ | 0.253 | 0.297 | 0.394 | - | - | - | **0.681** |

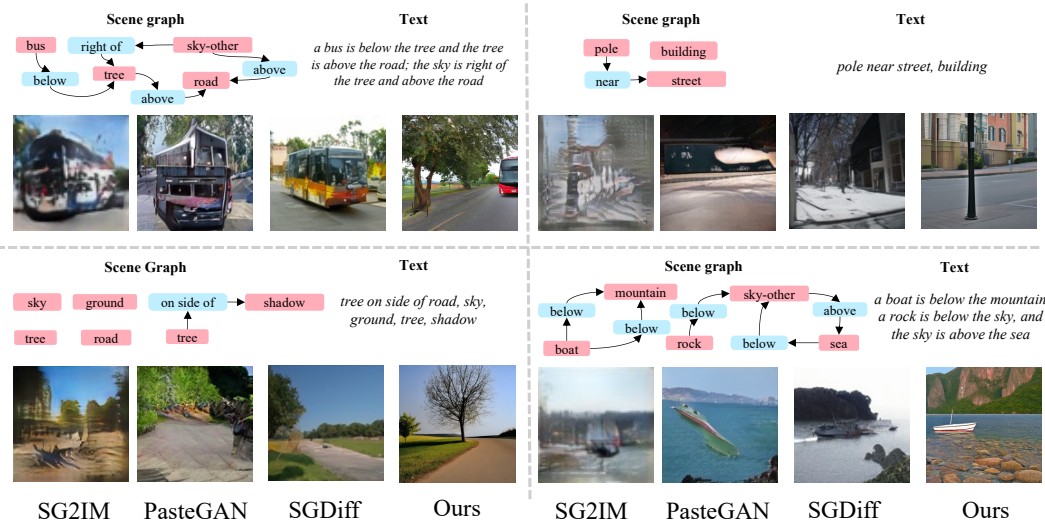

|        |          |        |      |
|--------|----------|--------|------|
| SG2IM  | PasteGAN | SGDiff | Ours |

Figure 4: **Comparison with Scene Graph to Image Generation**. SG-Adapter outperforms other SG generation methods in terms of image quality and relation accuracy.

For quantitative comparison, due to the large size of the validation sets in the above datasets, performing GPT-4V-based evaluations for all methods would require a significant number of API calls, exceeding our available resources. Therefore, we randomly sampled 200 samples to evaluate SG-IoU, Entity-IoU, and Relation-IoU. We provided the FID and Inception Score (Salimans et al., 2016) for each method in Tab. 3. The results were also tested on the COCO-Stuff validation dataset with a resolution of 256×256. Tab. 3 shows that SG-Adapters outperform other SG-to-Image approaches on both evaluation metrics by a significant margin, demonstrating superiority in both image fidelity and diversity.

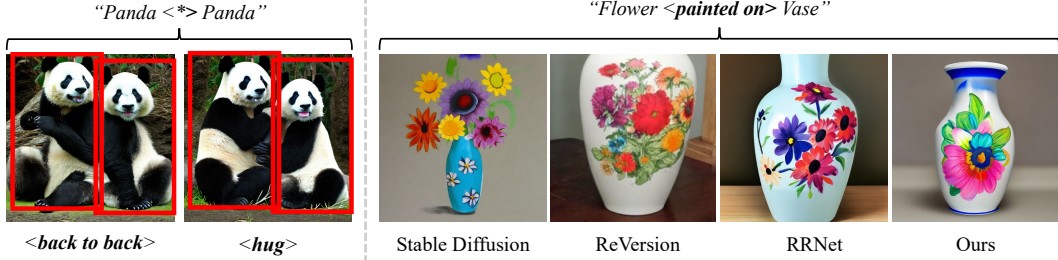

Figure 5: **Left:** The same layout appears visually different due to different relationships. **Right:** Our method is also capable of learning the customized single relationship.

## 5 CONCLUSION AND DISCUSSION

This paper introduces the Scene Graph Adapter (SG-Adapter), a novel enhancement for text-to-image generation models. By integrating scene graph knowledge, the SG-Adapter significantly improves the contextual understanding of these models, ensuring images closely match their textual descriptions.

The adapter employs an efficient triplet-token attention mechanism within a transformer architecture, allowing for more precise mapping of text to visual elements. To validate its effectiveness, the study utilized a specially curated dataset featuring multiple relations and high-quality annotations, showing the crucial role of clean, relation-rich data in multi-relational learning. As shown in Fig.5, SG-Adapter can also learn single complex relations as a by-product. Furthermore, we propose three effective metrics derived from GPT-4V (Achiam et al., 2023) to evaluate the generated relation precisely. Lastly, our method could learn a new and complex single relation as a by-product, similar to ReVersion (Huang et al., 2023) and RRNet (Wu et al., 2024), as demonstrated in Fig.5.

As for limitation, to address data privacy concerns and follow the double-blind policy, we applied anonymization techniques to human faces in our MultiRels dataset, which may introduce some artifacts affecting image quality inevitably. We are actively exploring more sophisticated anonymization methods that preserve data integrity while ensuring privacy.

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

# A APPENDIX

## A.1 INITIAL EXPERIMENT WITH SCENE GRAPH ATTENTION MASK

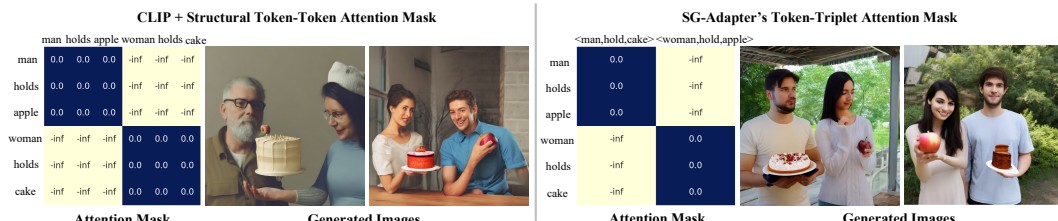

Figure 6: **Results of Initial Attempt and SG-Adapter.** Directly integrating the structural token-token attention mask in a hard way within the CLIP model could break the delicate balance of learned sequential dependencies and fail to ensure accurate semantic structure in the generated images. Instead, our SG-Adapter makes use of a novel token-triplet attention mask in a learnable way to correct the unreasonable token-token interactions and guarantee precise relation correspondence.

Our initial attempts involved the utilization of an adjusted attention mask $\mathbf{M}^\tau$ within the transformer's attention mechanism of the CLIP text encoder, aiming to ensure intra-triplet attention cohesion and maintain the semantic structure of caption. This integration, however, did not translate to empirical improvements as shown in Fig. 6. We observed that directly manipulating the attention mask of the pre-trained CLIP model could disrupt the delicate balance of learned sequential dependencies, adversely affecting the quality of text-to-image synthesis due to inconsistencies between the model's training and inference methodologies.

## A.2 ABLATION OF NUMBER OF PARAMETERS

## A.3 MULTIRELS BENCHMARK DETAILS

As mentioned in the main paper, we organized the MultiRels into two parts: Reversion and Multiple Relations. This section will introduce more detailed information about the Multiple Relations part.

The Multiple Relations part contains 210 samples. We initially plan to collect all the images from the Internet and then label them manually. However, after we had collected dozens of images, we found this way not efficient enough since there are very few images that contain multiple clear and salient relations. Besides, a text describing a multiple relations image tends to be long so the relevance of retrieved images will also decrease with the longer text. Therefore we only collect 40 multi-relations images from the Internet by retrieving a long text.

On the contrary, 170 images in the Multiple Relations part were taken by ourselves.

1. Concerning human relations, we mainly focus on "*holding*", "*drinking*", "*stands on*", "*sits on*" such kinds of simple human actions, combining random 2-3 relations from them to result in 15 different text templates containing multiple relations finally. To make the collected images as diverse as possible, we arranged for 5 volunteers to participate in the photoshoot in 6 different environmental backgrounds both indoor and outdoor. **We re-paint faces in these photos locally for the privacy of the volunteers and the blind paper review**. We examples of the above data in Fig.10, Fig.11.

2. For object relations, we consider the positional relationships like "*is above*", and "*is under*". We place various normal objects, e.g. fruits and daily necessities, on the same/different side of an object like a table, chair, bench, and so on. Also, for the diversity of the data set, we take photos of these objects in 6 different indoor and outdoor environments. There are examples provided in Fig.12, Fig.13.

The 15 templates we adopt in human relations are as follows:

1. *a man stands on floor and a woman sits on a chair*.
2. *a woman stands on floor and a man sits on a chair*.

3. *a man stands on floor and a woman stands on floor.*

4. *a man sits on a chair and a woman sits on a chair.*

5. *a man drinking milk and a woman drinking cola.*

6. *a man sits on a chair and a woman holding a ⟨obj⟩ stands on floor.*

7. *a woman sits on a chair and a man holding a ⟨obj⟩ stands on floor.*

8. *a man drinking juice sits on a chair and a woman stands on floor.*

9. *a man holding a ⟨obj⟩ stands on floor and a woman drinking water stands on floor.*

10. *a man stands on floor and a man sits on a chair.*

11. *a man drinking water sits on a chair and a man holding a ⟨obj⟩ stands on floor.*

12. *a man holding a ⟨obj⟩ stands on floor and a man sits on a chair.*

13. *a man holding a ⟨obj⟩ stands on floor and a man stands on floor.*

14. *a man drinking juice and a man drinking water.*

15. *a man drinking water and a man drinking water.*

To see the complete data set and its corresponding metadata, please refer to the compressed file MultiRels.zip.

## A.4   ADDITIONAL QUALITATIVE RESULTS

We present additional qualitative results of our SG-Adapter compared with other baseline methods in Fig.14, Fig.16.

## A.5   GPT-4V PROMPTS

**Prompt to Extract Scene Graph from Image:** *Please extract the scene graph of the given image. The scene graph just needs to include the relations of the salient objects and exclude the background. The scene graph should be a list of triplets like ["subject", "predicate", "object"].*
*Both subject and object should be selected from the following list: ["an astronaut", "a ball", "a cake", "a box", "a television", "a table", "a horse", "a pitaya", "a woman", "a book", "a laptop", "a bottle", "a banana", "a sofa", "floor", "water", "an apple", "a chair", "a pineapple", "an umbrella", "a boy", "a paper", "a bear", "a girl", "a panda", "a cup", "a man", "a bike", "a carrot", "a phone"].*
*The predicate should be selected from the following list: ["stands on", "is above", "drinking", "is under", "sits back to back with", "ride on", "holding", "sits on"]*
*Besides the scene graph, please also output the objects list in the image like ["object1", "object2", ..., "object"]. The object should be also selected from the above-mentioned object list. The output should only contain the scene graph and the object list.*

**Prompt to Parse Caption to Scene Graph:** *Here I have a group of captions and please help me to parse each one. Each caption should be transformed to a Scene Graph reasonably which is composed of some relations. A relation is a triplet like [subject, predicate, object] and please replace the original pronouns with reasonable nouns. Both subject and object should only have one object or person. "and" relation should not be included in the Scene Graph. Besides, I want to get the indexes of all of the subjects, predicates, and objects in the original caption, which is called mapping here. For example, the caption is: a boy holding a bottle shakes hands with a girl sitting on a bench.*
*The corresponding Scene Graph should be: [[a boy, holding, a bottle], [a boy, shakes hands with, a girl], [a girl, sitting on, a bench]].*
*The indexes of each word(punctuation is also considered a word) in the caption(called all_words_indexes) are:"a":0, "boy":1, "holding":2, "a" :3, "bottle":4, "shakes":5, "hands":6, "with":7, "a":8, "girl":9, "sitting":10, "on":11, "a":12, "bench":13.*
*The index of every word in the Scene Graph(i.e., the mapping) should be:["a":0, "boy":1, "holding":2," a":3, "bottle":4,"a":0, "boy":1, "shakes":5, "hands":6, "with":7, "a":8, "girl":9, "sitting":10, "on": 11,"a":12, "bench":13]*
*The indexes of all of the subject, predicate, and object in the original caption should be highly precise. The results of each caption are data like:*

*scene graph:*
*all_words_indexes:*
*mapping:*
*When generating the mapping please refer to the scene graph and the all_words_indexes to ensure the correct result.*
*The captions are:*

### A.6   IMPLEMENTATION DETAILS

All the experiments are conducted on $768 \times 768$ image resolution and all the models are trained on a single A100 GPU for several hours. We adopt SD 2.1 as our base model. We set training batch size 4, learning rate 1e-5. We adopt the optimizer AdamW and most models converge around 12000 to 14000 iterations. During the sampling process, we use a parameter $\tau$ to balance the ability to control and the image quality. For a diffusion process with $T$ steps, we could use scene graph guided inference at the first $\tau * T$ steps and use the standard inference for the remaining $(1 - \tau) * T$ steps. In this paper, $\tau$ set to 0.3 is enough to achieve relation control while maintaining the image quality.

### A.7   MORE ANALYSIS

We also provide additional results shown in Fig. 7. They demonstrates the SG-Adapter's effectiveness in addressing challenges related to generalization, attribute binding, and complex multi-entity interactions. The SG-Adapter consistently generates accurate and coherent outputs across diverse scenarios, including those not commonly observed in the training data. In the **Generalization Test**, it handles novel relationships such as a cat shaking hands with another cat and a hamster riding a motorbike, showing its ability to move beyond high-frequency training patterns. The **Attribute Binding Test** confirms its capability to assign correct attributes, such as maintaining consistent colors for objects like "a blue cat and a white dog." Additionally, in **More Entities**, the SG-Adapter successfully manages multi-entity scenes, such as "two astronauts standing while a man is sitting" and "three pandas sitting together," showcasing its adaptability to diverse and complex relationships. These results highlight the SG-Adapter's ability to generate meaningful and accurate outputs, demonstrating robustness and versatility across varied tasks.

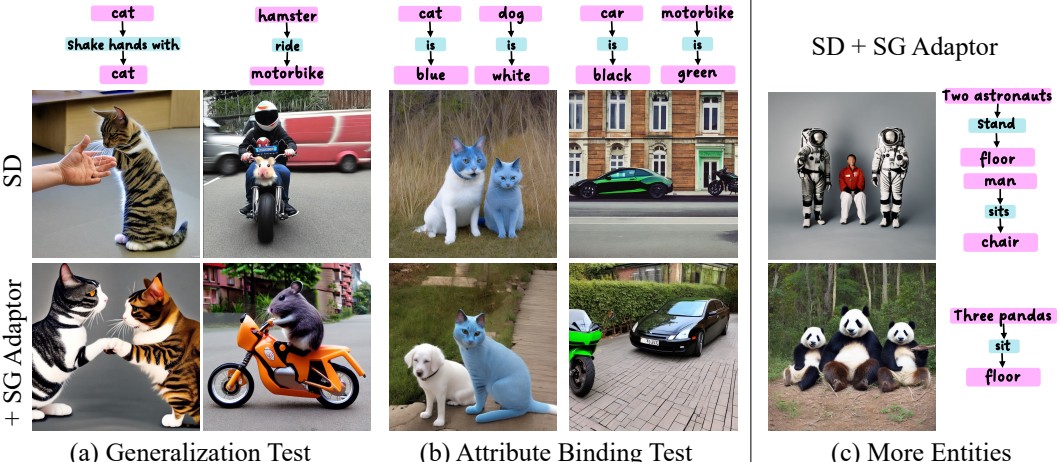

(a) Generalization Test          (b) Attribute Binding Test          (c) More Entities

Figure 7: **(a) Generalization Test**: Novel interactions beyond the training set, such as a cat shaking hands with another cat or a hamster riding a motorbike. **(b) Attribute Binding Test**: Resolves attribute binding, e.g., assigning correct colors to objects like "a blue cat and a white dog." **(c) More Entities**: Handles multi-entity scenes, such as "two astronauts standing while a man is sitting" or "three pandas sitting together."

Ours

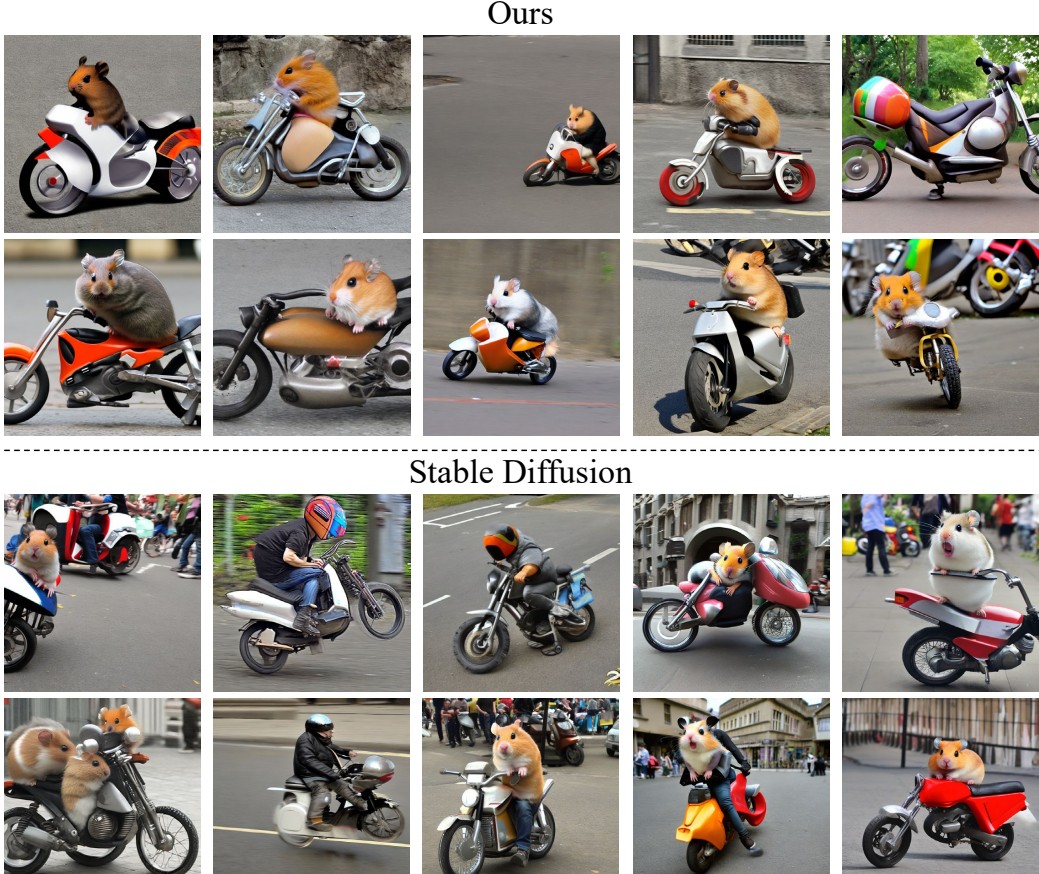

Figure 8: **Success Rate.**

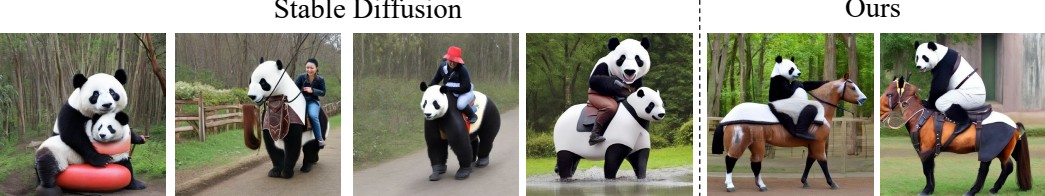

Figure 9: **Limitations.**

**text**:   *a man sits on a chair and a girl holding a phone stands on floor*

**index**:   1  2  3  4  5  6    7  8  9    10    11  12    13  14  15

**scene graph**:                                                      **mask mapping**:

[[a man, sits on, a chair],   ⟶   [[1, 2, 3, 4, 5, 6],

[a girl, holding, a phone],   ⟶   [8, 9, 10, 11, 12],

[a girl, stands on, floor]]   ⟶   [8, 9, 13, 14, 15]]

Figure 10: **Example 1 of MultiRels**

**text**:   *a man holding a carrot stands on floor and a man sits on a chair*

**index**:   1  2    3    4  5    6    7    8    9  10  11    12  13  14  15

**scene graph**:                                                      **mask mapping**:

[[a man, holding, a carrot],   ⟶   [[1, 2, 3, 4, 5],

[a man, stands on, floor],   ⟶   [1, 2, 6, 7, 8],

[a man, sits on, a chair]]   ⟶   [10, 11, 12, 13, 14, 15]]

Figure 11: **Example 2 of MultiRels**

**text**:   *a pineapple is above a table and a bottle is under the table*

**index**:   1    2    3  4  5  6    7  8  9  10  11  12  13

**scene graph**:                                                      **mask mapping**:

[[a pineapple, is above, a table],   ⟶   [[1, 2, 3, 4, 5, 6],

[a bottle, is under, the table]]   ⟶   [8, 9, 10, 11, 12, 13]]

Figure 12: **Example 3 of MultiRels**

**text**:   *a bottle is above a bench and an apple is under the bench*

**index**:   1  2  3  4  5  6    7  8  9  10  11  12  13

**scene graph**:                                                      **mask mapping**:

[[a bottle, is above, a bench],   ⟶   [[1, 2, 3, 4, 5, 6],

[an apple, is under, the bench]],   ⟶   [8, 9, 10, 11, 12, 13]]

Figure 13: **Example 4 of MultiRels**

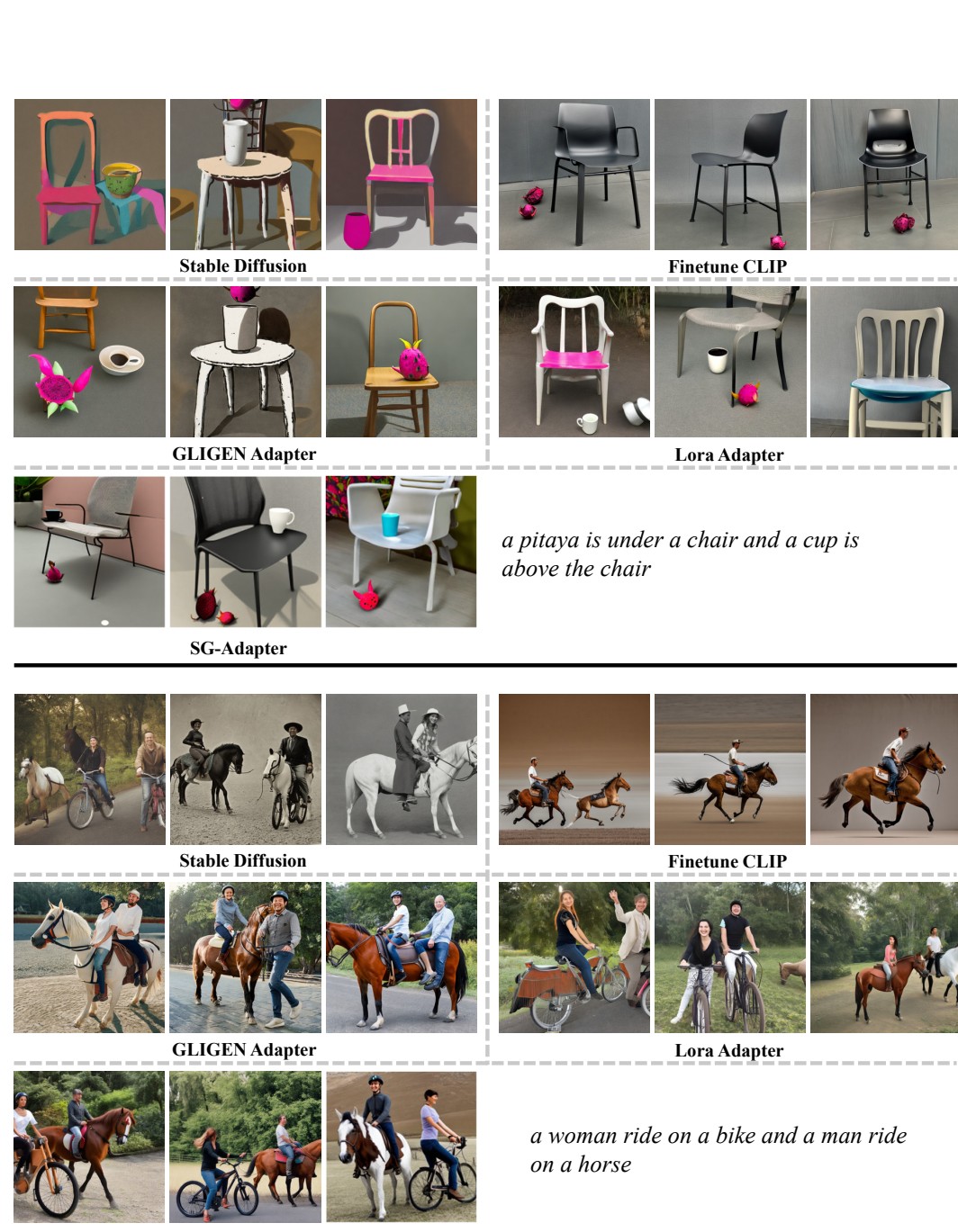

Figure 14: **More qualitative results-1.**

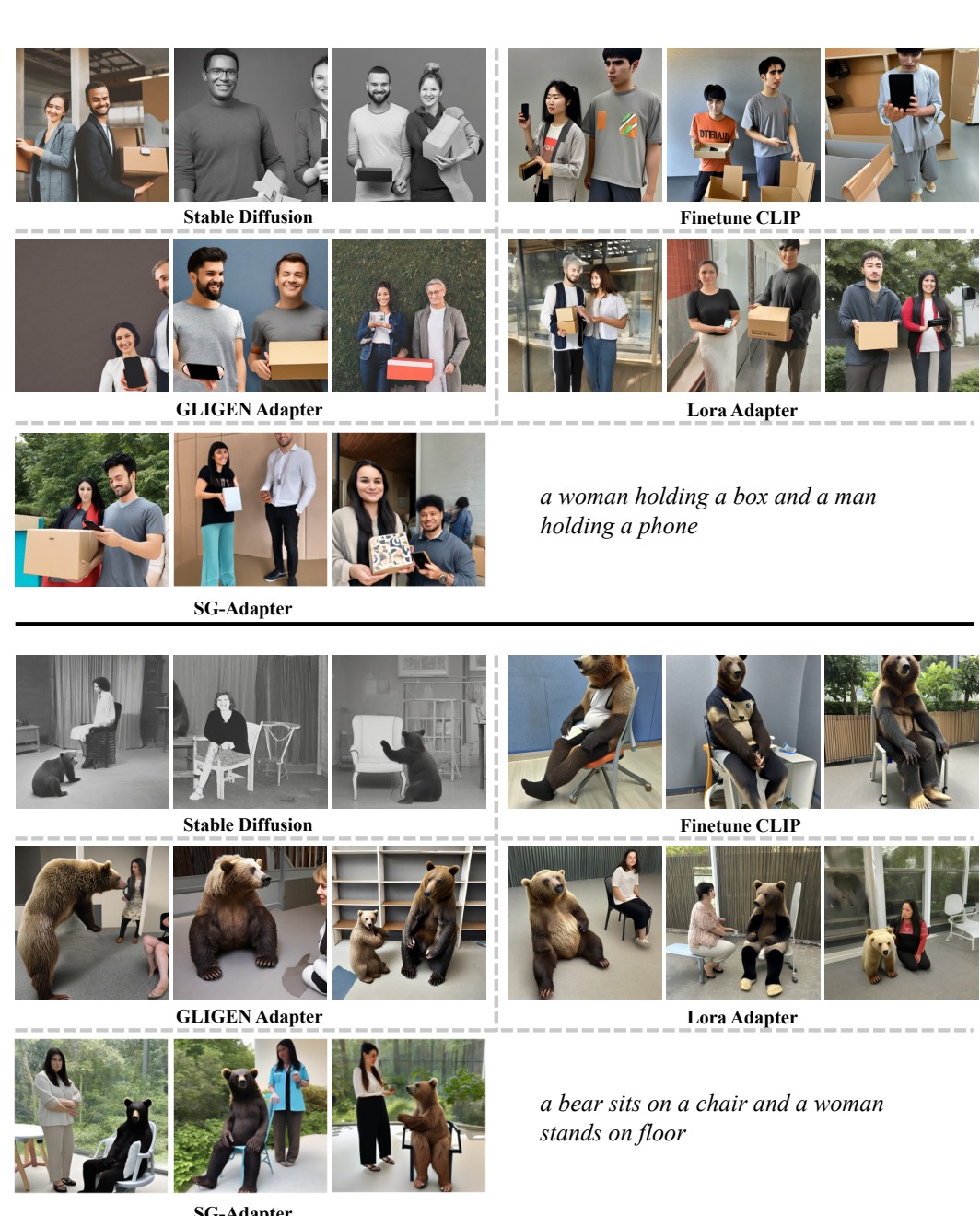

*a woman holding a box and a man holding a phone*

*a bear sits on a chair and a woman stands on floor*

Figure 15: **More qualitative results-2.**

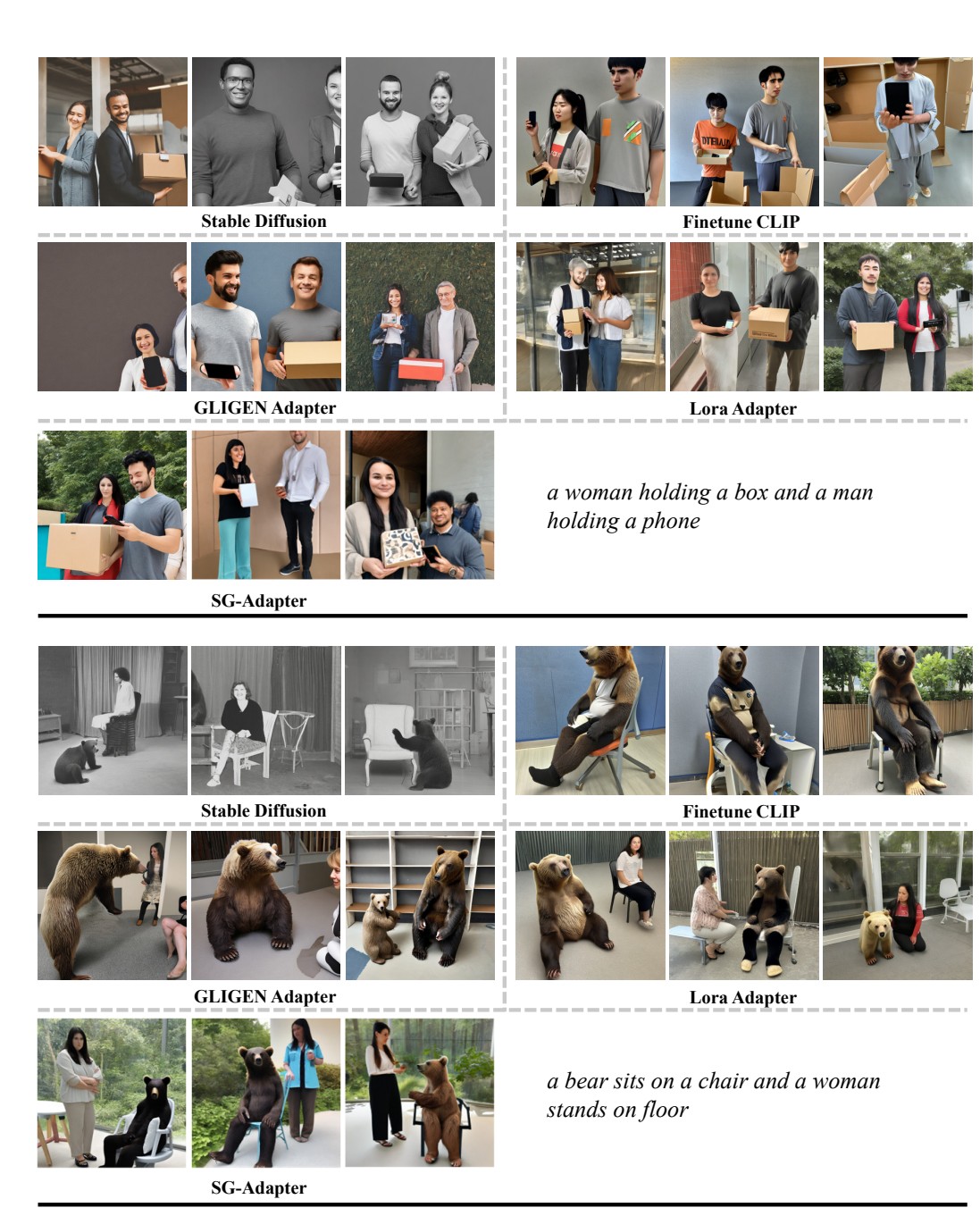

Stable Diffusion

Finetune CLIP

GLIGEN Adapter

Lora Adapter

SG-Adapter

*a woman holding a box and a man holding a phone*

Stable Diffusion

Finetune CLIP

GLIGEN Adapter

Lora Adapter

SG-Adapter

*a bear sits on a chair and a woman stands on floor*

Figure 16: **More qualitative results-2.**

