# OpenReview forum: "SG-Adapter: Enhancing Text-to-Image Generation with Scene Graph Guidance"
_ICLR.cc/2025/Conference — Submitted to ICLR 2025_

### Official Review · Reviewer_wQ1r · 2024-11-02

**Soundness:** 3
**Presentation:** 2
**Contribution:** 2
**Rating:** 5
**Confidence:** 4

**Summary:**

This paper presents a Scene Graph Adapter for text-to-image generation that leverages scene graph to refine CLIP text embeddings. Additionally, a dataset of 309 images with relational annotations is proposed for evaluation.

**Strengths:**

1. The paper contributes a scene graph dataset with relational annotations, which adds value to the community.
2. The authors include their code in the supplemental materials.
3. Human evaluations are conducted to assess the effectiveness of the proposed method.

**Weaknesses:**

1. Experiments are conducted only on the proposed dataset, which is relatively small and may not provide a robust evaluation.
2. The method uses a Scene Graph to refine CLIP text embeddings, which may be overly simplistic. I think the adapter should also change the parameters of the Stable Diffusion.
3. The latest method compared in Table 1 is from 2023. Including more recent methods would strengthen the comparative analysis.
4. The authors note that fine-tuning Stable Diffusion on a small dataset can degrade FID (shown in Table 1). Fine-tuning on a larger dataset, such as Flickr30k Scene Graph, is recommended.
5. Human evaluation is conducted on only 20 cases (from the proposed test set). Expanding the test set could improve the reliability of these results.

**Questions:**

See the detail in the weaknesses

---

> ### Author Response · Authors · 2024-11-21
>
> Thank you for your thoughtful and constructive feedback. Your comments have been invaluable in helping us refine and improve our study. Below, we address your concerns in detail.
>
> ---
>
> ## Weaknesses
>
> ### Weakness 1, 3, 4: Dataset Size and Comparative Evaluation
> **Response**: Thank you for your suggestions. In addition to experiments conducted on our MultiRels dataset, we have conducted extensive evaluations on Flickr30k[1] to demonstrate the robustness and generalization performance of our method. Please refer to Section 4.6 (L420-L465), Table 3, and Figure 4 for these results.
>
> During the rebuttal period, **we have additionally included R3CD (AAAI24) and ISGC (24)** in Table 3 to provide a more comprehensive comparative analysis. The updated results are as follows:
>
> | **Method**              | **SG2IM[2]** | **PasteGAN[3]** | **SGDiff[4]** | **SceneGenie[5]** | **R3CD (AAAI24)[6]** | **ISGC (24)[7]** | **SG-Adapter** |
> |--------------------------|-----------|--------------|------------|----------------|-------------------|---------------|----------------|
> | **FID**$\downarrow$    | 99.1      | 79.1         | 36.2       | 62.4           | 32.9              | 38.1          | **25.1**       |
> | **Inception Score**$\uparrow$ | 8.20      | 12.3         | 17.8       | 21.5           | 19.5              | 30.2          | **57.8**       |
> | **SG-IoU**$\uparrow$   | 0.085     | 0.091        | 0.122      | -              | -                 | -             | **0.413**      |
> | **Entity-IoU**$\uparrow$ | 0.297     | 0.382        | 0.436      | -              | -                 | -             | **0.729**      |
> | **Relation-IoU**$\uparrow$ | 0.253     | 0.297        | 0.394      | -              | -                 | -             | **0.681**      |
>
> For R3CD, ISGC, and SceneGenie, we report inception scores and FID values directly from their respective papers, as their code is not available. These additional results help establish a more robust and fair comparison.
>
> ---
>
> ### Weakness 2: Adaptor vs. Stable Diffusion Fine-tuning
> **Response**: Our analysis identified the primary source of failure as the causal attention process in CLIP. Consequently, our focus has been on rectifying this issue.
>
> We also conducted experiments where only the parameters of the Stable Diffusion (SD) UNet were fine-tuned. However, this yielded suboptimal results, reinforcing the importance of addressing causal attention in the CLIP text embeddings. This focus ensures that our approach remains efficient and targeted, avoiding unnecessary changes to the Stable Diffusion backbone.
>
> ---
>
> ### Weakness 5: Human Evaluation Sample Size
> **Response**: Compared to previous works, our method collects human evaluation data on a much larger scale. For instance:
> - **Attend\&Excite[8]** gathered 65 responses across 10 prompts using 4 methods, resulting in **2,600 interactions**.
> - **Dreambooth[9]** collected **1,800 interactions** with 72 responses and 25 prompts.
>
> In contrast, our study involved **104 responses for each of 20 text prompts across 5 comparison methods**, yielding a total of **10,400 interactions**. This expanded dataset significantly enhances the reliability and robustness of our evaluation, providing a strong foundation for our conclusions.
>
> We acknowledge the importance of expanding the evaluation set further and will consider additional test cases in future work to address more diverse scenarios comprehensively.
>
> ---
>
> We hope these detailed clarifications and additional experimental results address your concerns thoroughly. Thank you again for your valuable feedback, which has greatly helped us strengthen our work.
>
> [1] Flickr30k entities: Collecting region-to-phrase correspondences for richer image-to-sentence model.
> [2] Image generation from scene graphs.
> [3] Pastegan: A semiparametric method to generate image from scene graph.
> [4] Diffusion-based scene graph to image generation with masked contrastive pre-training.
> [5] Scene graph to image synthesis via knowledge consensus.
> [6] R3cd: Scene graph to image generation with relation-aware compositional contrastive control diffusion.
> [7] Scene graph to image synthesis: Integrating clip guidance with graph conditioning in diffusion models.
> [8] Attend-and-Excite: Attention-Based Semantic Guidance for Text-to-Image Diffusion Models.
> [9] DreamBooth: Fine Tuning Text-to-Image Diffusion Models for Subject-Driven Generation

---

> > ### Author Response · Authors · 2024-11-24
> >
> > Dear Reviewer,
> >
> > Thank you again for your valuable feedback and thoughtful comments during the discussion phase. We would like to kindly remind you that the discussion period will conclude on **November 26th**. If you have any additional questions, concerns, or clarifications you would like us to address, we would be more than happy to provide prompt responses. Your insights have been instrumental in shaping the final version of our submission, and we greatly appreciate your time and effort in engaging with our work.
> >
> > Thank you for your attention, and we look forward to hearing from you!

---

> ### Author Response · Authors · 2024-12-02
> **Authors' Kind Reminder to Reviewer wQ1r**
>
> Dear Reviewer wQ1r,
>
> We sincerely appreciate your time and effort in reviewing our submission and writing your responses.
>
> **However, as the discussion deadline approaches in less than 2 days (only 1 day left for reviewers to post a message), we would greatly appreciate it if you could carefully review our latest responses and provide further clarifications at your earliest convenience. We highly value any opportunities to address any potential remaining issues, which we believe will be helpful for a higher rating. Your timely feedback is extremely important to us!**
>
> Once again, thanks so much for your time and effort, we deeply appreciate it!
>
> Best Regards,
>
> Authors of Submission 1595

---

### Official Review · Reviewer_Bw61 · 2024-11-03

**Soundness:** 3
**Presentation:** 3
**Contribution:** 2
**Rating:** 5
**Confidence:** 4

**Summary:**

This paper hopes to optimize the generation of entity interaction in text-to-image generation by introducing the scene graph structure into the adapter. The method is simple and clear and has achieved significant improvement compared to the baseline.

**Strengths:**

- The motivation is straightforward. The description of the problem and method is simple, clear and understandable.
 - Entity interaction information is introduced through scene graph to optimize the generation of entity interaction, which is relatively direct.
 - Judging from the demo and numerical comparison results, there are good results.

**Weaknesses:**

- The novelty in method design is limited. It only performs an attention on the text embedding based on the scene graph relationship. It can even be considered as a replacement of the key and value values in the cross attention structure. Simplicity is not a disadvantage. If it is simple, direct and works, the author can provide some discussions to analyze the most important factors behind it.
Is the adapter structure important or is the scene graph information the most important? For example, in addition to the adapter, can simply injecting scene graph information through cross attention also obtain similar results?
And whether there are some situations where scene graph information is not applicable.
 - The experimental evaluation of entity interaction generation is not comprehensive enough. The scene graph detection method itself will also have a very obvious long-tail phenomenon for entity relationships. For entity interaction image generation, this should also be considered. Distinguishing and explaining according to the detection frequency in the training dataset will make the evaluation of the model more comprehensive and convincing. The author can try to divide buckets according to the frequency of scene graph detection, and finally give the results of each bucket and the average result.

**Questions:**

-  The currently shown results (Figure 3) seem to have at most two main bodies interacting with their accessories. 1) In the case of more main bodies, such as a crowd surrounding or three people facing away from each other; 2) For some pairs that may not be common in the training set, such as pandas riding on horsebacks, what is the generation effect like? This helps evaluate whether the sg-adapter has learned to generate general interaction relationships or just high-frequency combinations in the training set.
 - The effect of scene graph detection may not be 100% accurate. What will be the effect when the detection result conflicts with the meaning of the text itself? Will it be more serious for some freely input long texts? For example, when users use the text-to-image function, they often add a long string of magic prompts, such as 4K, realistic, specific lenses, etc. These may all have an impact on scene graph detection.
 - How many parameters need to be fine-tuned in the current adapter part? Will the size of the number of parameters affect the final effect? The analysis of this part may make the article more complete.

In general, I think the problem solved by this paper is quite important. However, I feel that the current method is not general enough and may be a temporary technique. When there is a specific need for interaction generation in a specific scene, fine-tuning based on the SG Adapter will have some improvement. But it does not fundamentally solve the problem of entity interaction. Because I think the introduction of the SG adapter will lead to the problem that entity interaction adds the long-tail problem of scene graph detection on top of the long-tail problem of interaction relationships in the training set itself. The impact on the community may be limited. If the author has more thoughts and ideas on this aspect, we can discuss them.

---

> ### Author Response · Authors · 2024-11-21
>
> We appreciate your recognition and valuable feedback on our work. Your comments are insightful and have helped us refine and improve our study. Below, we address your concerns and suggestions in detail.
>
> ## Weaknesses
>
> ### Weakness 1: Limited Novelty in Method Design
> 1) **The Importance of Scene Graph Information**:
>    Our analysis highlights that incorporating scene graph information is essential for rectifying false contextualization caused by erroneous causal attention. Furthermore, how scene graph information is utilized significantly impacts performance.
>
> 2) **Utilizing Scene Graph Information Effectively**:
>    We explored alternative methods, such as inserting scene graph embeddings via gated self-attention. However, this approach failed to effectively address the errors, as shown in the third row of Table 1 in the main paper. In contrast, our adaptor demonstrates superior correction capabilities, as validated by the experimental results. This demonstrates the importance of both the adaptor structure and the specific way in which scene graph information is incorporated.
>
> ### Weakness 2: Experimental Evaluation of Entity Interaction Generation is Not Comprehensive
> Instead of relying on traditional scene graph generation methods, which are often trained on limited datasets and prone to long-tail distribution issues, we leveraged GPT-4V to extract scene graphs. Recent research ([1], [2]) combining large language models (LLMs) with scene graphs demonstrates that LLM-based methods, due to their strong generalization capabilities, are less affected by the long-tail distribution problem.
>
> - **References**:
>   1. LLM4SGG: Large Language Models for Weakly Supervised Scene Graph Generation
>   2. GPT4SGG: Synthesizing Scene Graphs from Holistic and Region-Specific Narratives
>
> ---
>
> ## Questions
>
> ### Question 1: Generating Complex Interactions
> To address this concern, we have included additional results in Appendix A.7 Fig.7, covering cases with more complex interactions. For example:
> - **Crowded scenarios**, such as a group of astronauts.
> - **Uncommon pairs**, such as "hamster riding on motorbike."
>
> These cases demonstrate that our SG-adaptor effectively learns general interaction relationships beyond high-frequency combinations in the training set.
>
> ### Question 2: Scene Graph Detection Conflicts with Text Prompts
> Our framework focuses exclusively on entities and their relationships. For unrelated prompt elements such as "4K," "realistic," or specific lens types, we provide the flexibility to process text using either GPT-4 or an NLP parser. For handling lengthy or complex prompts, GPT-4 can be used with additional instructions such as:
> - *"Ignore prompts unrelated to entities and relationships."*
>
> This ensures that the scene graph detection process remains focused on the core elements relevant to the task, mitigating potential conflicts with extraneous text.
>
> ### Question 3: Impact of Adaptor Parameter Size
> We have conducted an analysis of how performance varies with the size of the parameters in the adaptor. Detailed results are provided in the following table.
>
> | number of parameters | SG-IoU | Entity-IoU | Relation-IoU | FID |
> |----------------------|--------|------------|--------------|-----|
> | 8.666M               | 0.554  | 0.805      | 0.748        | 26.0|
> | 11.55M our setting   | 0.623  | 0.812      | 0.753        | 26.2|
> | 17.85M               | 0.628  | 0.803      | 0.766        | 28.2|
> | 24.15M               | 0.630  | 0.819      | 0.755        | 30.5|
>
> The findings show that as the number of parameters increases:
> - **FID** worsens (higher values).
> - **Alignment with Scene Graph** slightly improves.
>
> We have chosen a balance point between FID and scene graph alignment to ensure optimal performance without over-parameterization.
>
> ---
>
> We hope these clarifications and additional experiments address your concerns. Thank you for your thoughtful feedback, which has helped us strengthen our work.

---

> > ### Comment · Reviewer_Bw61 · 2024-11-23
> >
> > Thank you for your response, which has addressed most of my concerns. I still have a question regarding the generation results. In the appendix, you provided some new generation results. What is the success rate of generating these subject interactions? What are the limitations of SGAdapter? I noticed that you did not generate the example I mentioned, "pandas riding on horsebacks" (which is understandable, even if the results are not perfect). I am just curious about the success rate of generating reasonably good results, as this seems quite crucial.

---

> > > ### Author Response · Authors · 2024-11-23
> > >
> > > We sincerely appreciate your thoughtful feedback and detailed questions. Your insights have been invaluable in helping us refine our explanation and further evaluate our method's performance. **We have updated our manuscript** to provide two additional clarifications:
> > >
> > > ---
> > >
> > > ### Success Rate
> > >
> > > - **Qualitative Demonstration:** To visually assess the success rate, we generated 10 images using different random seeds for the same text prompt with both Stable Diffusion and our method. As shown in the **Appendix.Figure 8**, our method exhibits a higher likelihood of generating correct images.
> > >
> > > - **Quantitative Demonstration:** The metrics presented in Table 3 of the main paper further validate this observation. Specifically, the scene graph IoU metric provides a quantitative measure of how well the generated images align with the given scene graphs, offering a clear indication of the success rate.
> > >
> > > ---
> > >
> > > ### Limitations
> > >
> > > **We have included the example of "pandas riding on horsebacks" in the Appendix Figure.9**. Our SG-Adapter could improve the Stable Diffusion and generate resonable images, but the success rate for this prompt is relatively low. This is primarily because such scenarios are not represented in any training dataset, requiring to rely entirely on model's generalization capabilities. For comparison, we also included results generated directly by Stable Diffusion, **as shown in the Appendix Figure.9 Left**, which demonstrate an almost complete inability to produce correct outputs.
> > >
> > > This phenomenon highlights a characteristic of our approach: as our method builds on a pre-trained base model rather than being trained from scratch, its performance naturally reflects the generative capabilities of the underlying model.
> > >
> > > Looking forward, we plan to integrate our method with state-of-the-art generative models, such as Flux. By leveraging these advanced models, we believe our approach can further extend its capabilities to handle more complex scenarios and achieve even better performance on challenging cases.

---

> > > > ### Comment · Reviewer_Bw61 · 2024-11-25
> > > >
> > > > Thank you very much for the prompt response.
> > > >
> > > > From the newly provided results, it appears that the proposed method shows some improvement in text response compared to the baseline (original SD).
> > > >
> > > > From Figure 8, under a loose standard, the success rate difference might be around 10% (from 80% to 90%). Under a relatively moderate standard, the proposed method might show some improvement (from 30% to 80%). Under a stricter standard, both the baseline and proposed method are unusable with obvious physical rule errors. This may be very subjective and have a large variance. Based on the currently provided images, subjectively, the improvement seems incremental.
> > > >
> > > > Compared to SD, the improvement after fine-tuning is expected, but the success rate improvement should be analyzed more systematically to better demonstrate the method's effectiveness. Currently, there is only one case and 20 images, which makes it hard to consider this a solid investigation. The discussion on limitations is similar, as it seems to involve only one image. I believe this paper has its merits, but the current results make it difficult for me to consider it up to ICLR standards. Therefore, I am inclined to maintain the current rating and suggest the authors conduct a more detailed analysis of the generated results.

---

> > > > > ### Author Response · Authors · 2024-11-25
> > > > >
> > > > > Thank you very much for your thoughtful response and constructive feedback.
> > > > >
> > > > > We appreciate your acknowledgment of the improvement our method demonstrates compared to the baseline (SD).
> > > > > We believe that both you and we agree that **visually demonstrating the success rate based on a small number of cases is inherently subjective and may lack statistical significance**  for the following reasons:
> > > > > - Individual user preferences can be highly subjective and prone to bias.
> > > > > - The sample size is too small to draw reliable statistical conclusions.
> > > > >
> > > > > The examples we provided in Figure 8 are meant as a **supplementary**  visual aid during rebuttal to help reviewers intuitively observe the improvements brought by our method. As you noted, our results show noticeable improvements over SD, even under consistent random seeds and identical hyperparameter settings. However, **we do not intend to use these limited examples as the primary evaluation metric for success rate**.
> > > > >
> > > > > Instead, in the main paper, we have already provided a systematic and quantitative evaluation to demonstrate the effectiveness of our method. Specifically:
> > > > > 1. **Objective Evaluation**: Success rate metrics from loose(Entity-IoU and Relation-IoU) to strict(SG-IoU) were evaluated using the most advanced GPT-4V as automatic evaluations. As demonstrated in Table.1 and Table.3, SG-Adapter outperforms other baseline methods significantly among all the Success Rate Metrics. **These results are evaluated across 220 test prompts with 10 seeds each among ours and baselines, resulting in 27,000 responses(9000 images x 3 metrics) from GPT-4V**.
> > > > >
> > > > > 2. **User Study**: To avoid the subjectivity of single-person-evaluation, **we conducted a user study involving 104 participants who were tasked with evaluating 20 questions, yielding a total of 10,400 interactions**. You can refer to Table.1 for details. This larger-scale evaluation mitigates the subjectivity associated with individual user assessments.
> > > > > 3. **Benchmark Dataset**: We tested our method on the large-scale scene graph dataset Flickr30k, further validating its performance.
> > > > >
> > > > > These evaluations offer a comprehensive and systematic reflection of the success rate improvements brought by our method, addressing both subjective and objective aspects of performance.
> > > > >
> > > > > Regarding the reviewer’s observation about "obvious physical rule errors" in the examples, we have already acknowledged these as limitations of our method. Specifically, such errors stem from the base model, which inherently has a probability of generating failure cases. Extending our approach to more advanced base models is an important direction for our future work and is expected to alleviate these limitations.
> > > > >
> > > > > We value your feedback and will strive to improve the clarity and comprehensiveness of our evaluations and discussions in future iterations.

---

> ### Author Response · Authors · 2024-12-02
> **Authors' Kind Reminder to Reviewer Bw61**
>
> Dear Reviewer Bw61,
>
> We sincerely appreciate your time and effort in reviewing our submission and writing your responses.
>
> **However, as the discussion deadline approaches in less than 2 days (only 1 day left for reviewers to post a message), we would greatly appreciate it if you could carefully review our latest responses and provide further clarifications at your earliest convenience. We highly value any opportunities to address any potential remaining issues, which we believe will be helpful for a higher rating. Your timely feedback is extremely important to us!**
>
> Once again, thanks so much for your time and effort, we deeply appreciate it!
>
> Best Regards,
>
> Authors of Submission 1595

---

> > ### Comment · Reviewer_Bw61 · 2024-12-03
> >
> > Thank you for the responses. After considering the other reviewers' opinions and the rebuttal feedback, I have decided to maintain my current score.

---

### Official Review · Reviewer_3Tge · 2024-11-04

**Soundness:** 3
**Presentation:** 4
**Contribution:** 3
**Rating:** 6
**Confidence:** 4

**Summary:**

To correctly align text-image consistency for text-to-image task, especially in complex scenarios involving multiple objects and relationships, this paper proposes SG-Adapter to leverage the structured representation of scene graphs to rectify inaccuracies in the original text embeddings. A highly clean, multi-relational scene graph-image paired dataset MultiRels is constructed to address the challenges posed by low-quality annotated datasets. Qualitative and quantitative experiments validate the efficacy of the proposed method in controlling the correspondence in multiple relationships.

**Strengths:**

a. The proposed SG-adapter is effective in correcting the incorrect contextualization in text embeddings and enhancing the structural
semantics generation capabilities of current text-to-image models.

b. Both qualitative and quantitative experiments demonstrate SG-Adapter outperforms compared SOTA methods.

**Weaknesses:**

a. The format of citations in this paper need be rectified.

b. This method requires the construction of a high-quality dataset, and it is mainly effective for the relationships within the constructed dataset, which will limit its generalization ability.

c. The quantitative results are insufficient and the author should provide more results to prove the its effectiveness.

**Questions:**

Can SG-Adapter solves the color bind problems?

---

> ### Author Response · Authors · 2024-11-21
>
> We appreciate your recognition and valuable feedback on our work. We apologize for the issue of the citation format and have fixed it in the updated manuscript. Your suggestions on our dataset and experiment are helpful in improving our work. Detailed responses are provided as follows:
>
> ### Weaknesses.a
> **Response**: Thanks for your comment we have rectified the format of citations in our manuscript.
>
> ### Weaknesses.b
> **Response**: We construct such a high-quality dataset to intuitively demonstrate the effectiveness of our method in generating multiple relationships with accurate correspondences. The relation combinations in our experiment part do not appear in our training dataset, showing that our method has a certain generalization ability. Additionally, as we illustrated in Section.4.6, **our method could also be extended to large-scale datasets like Flickr30k[1] and show consistent improvement**.
>
> ### Weaknesses.c
> **Response**: Thanks for your valuable suggestion. Besides the results on our MultiRels, we have included experimental results that were trained on the large-scale dataset Flickr30k. You can see Section.4.6 L420-L465, Tab.3 and Fig.4 for reference. During the rebuttal period, **we provided additional results in Appendix A.7** to show our model's generalization performance and ability to generate multiple entities. To better show the effectiveness of our method, we extended Tab.3 as follows:
>
> | **Method** | **SG2IM[2]** | **PasteGAN[3]** | **SGDiff[4]** | **SceneGenie[5]** | **R3CD(AAAI24)[6]** | **ISGC(24)[7]** | **SG-Adapter** |
> |------------|--------------|-----------------|---------------|-------------------|---------------------|-----------------|----------------|
> | **FID** $\downarrow$ | 99.1 | 79.1 | 36.2 | 62.4 | 32.9 | 38.1 | **25.1** |
> | **Inception Score** $\uparrow$ | 8.20 | 12.3 | 17.8 | 21.5 | 19.5 | 30.2 | **57.8** |
> | **SG-IoU** $\uparrow$ | 0.085 | 0.091 | 0.122 | - | - | - | **0.413** |
> | **Entity-IoU** $\uparrow$ | 0.297 | 0.382 | 0.436 | - | - | - | **0.729** |
> | **Relation-IoU** $\uparrow$ | 0.253 | 0.297 | 0.394 | - | - | - | **0.681** |
>
> For R3CD, ISGC and SceneGenie, we just report the inception score and FID from their paper since they do not provide the code.
>
> ### Questions.1
> **Response**: Yes. The SG-Adapter could alleviate the color leakage problem since we could regard color as a specific relationship, e.g., the caption "a blue cat" could correspond to the relation ["a cat", "is", "blue"]. We have provided this case in **Appendix A.7**.
>
> ---
>
> We hope these clarifications and additional experiments address your concerns. Thank you for your thoughtful feedback, which has helped us strengthen our work.
>
> [1] Flickr30k entities: Collecting region-to-phrase correspondences for richer image-to-sentence model.
> [2] Image generation from scene graphs.
> [3] Pastegan: A semiparametric method to generate image from scene graph.
> [4] Diffusion-based scene graph to image generation with masked contrastive pre-training.
> [5] Scene graph to image synthesis via knowledge consensus.
> [6] R3cd: Scene graph to image generation with relation-aware compositional contrastive control diffusion.
> [7] Scene graph to image synthesis: Integrating clip guidance with graph conditioning in diffusion models.

---

> > ### Author Response · Authors · 2024-11-24
> >
> > Dear Reviewer,
> >
> > Thank you again for your valuable feedback and thoughtful comments during the discussion phase. We would like to kindly remind you that the discussion period will conclude on **November 26th**. If you have any additional questions, concerns, or clarifications you would like us to address, we would be more than happy to provide prompt responses. Your insights have been instrumental in shaping the final version of our submission, and we greatly appreciate your time and effort in engaging with our work.
> >
> > Thank you for your attention, and we look forward to hearing from you!

---

> ### Author Response · Authors · 2024-12-02
> **Authors' Kind Reminder to Reviewer 3Tge**
>
> Dear Reviewer 3Tge,
>
> We sincerely appreciate your time and effort in reviewing our submission and writing your responses.
>
> **However, as the discussion deadline approaches in less than 2 days (only 1 day left for reviewers to post a message), we would greatly appreciate it if you could carefully review our latest responses and provide further clarifications at your earliest convenience. We highly value any opportunities to address any potential remaining issues, which we believe will be helpful for a higher rating. Your timely feedback is extremely important to us!**
>
> Once again, thanks so much for your time and effort, we deeply appreciate it!
>
> Best Regards,
>
> Authors of Submission 1595

---

> > ### Comment · Reviewer_3Tge · 2024-12-03
> >
> > Thank you for your efforts in addressing my questions and resolving my concerns. I maintain my score unchanged.

---

### Official Review · Reviewer_EbLH · 2024-11-04

**Soundness:** 3
**Presentation:** 1
**Contribution:** 3
**Rating:** 6
**Confidence:** 4

**Summary:**

This paper proposes SGAdapter which solves the problem of using sequential representations for input text in recent text-to-image generation methods. They augment their approach with scene-graph-to-image generation due to the graph representations. SGAdapter introduces a small-scale adapter between StableDiffusion and the CLIP encoder for this task. They also identify the problem of casual attention masks for scene graphs. They curate a new dataset for scene-graph-to-image generation. The experiments show that SGAdapter can generate realistic images from more complicated input sentences than baselines.

**Strengths:**

1. The method is simple and straightforward in solving the identified problem.
2. The empirical improvement is significant.

**Weaknesses:**

1. Some figures are not clear.
2. Some confusing notations.

**Questions:**

1. L40 missing space between "CLIP model" and the citation.
2. The $\mathbf{M}^{\tau}$ in Eq. 3 can be named differently, e.g. triplet-triplet mask, to distinguish it apart from the token-triplet attention mask in Eq. 5.
3. For Fig. 2, I suggest putting some of the introduced notations in Fig. 2 to help the reader understand the paper, e.g. model $f$, embedding $\mathbf{e}, \mathbf{w}$, etc. Fig. 2 can also be improved by using more arrows between features or modules to make it more concise and clear. For example, the inputs and outputs on the right side of the figure should be indicated with arrows. Redundant word in L233 Eq (equation) 4. The visualization of the matrix and the given scene graph example do not match. There are 2 triplets with 6 tokens in the input scene graph. I think it is a good opportunity to show what is $\mathbf{M}^{sg}$ and $\mathbf{M}^{\tau}$ for the inputs here.
4. L267, $\mathbf{M}$ or $M$?
5. In Eq. 5, what is the difference between $\mathbf{M}^{sg}$ or $M$?
5. I am confused about Tab. 2 and L413, is it about $\mathbf{M}^{\tau}$ or token-triplrt mask $\mathbf{M}^{sg}$. I thought $\mathbf{M}^{\tau}$ is for solving the problem of causal masks.
6. Fig. 1, I assume the add signs mean StableDiffusion + SGAdapter and Caption + Scene graphs. I think it can made clearer as initially I was not sure whether you were replacing captions with scene graphs or using both.

---

> ### Author Response · Authors · 2024-11-21
>
> We appreciate your commendation and valuable feedback on our work. We apologize for any confusion caused by the figures and notations. Below are the clarifications for your questions, and we have updated the corresponding parts in the manuscript accordingly:
>
> ### Questions 1-4, 7
> **Response**: We sincerely appreciate your suggestions regarding naming, formatting, and figure optimization. These improvements have been incorporated into the revised manuscript for your reference.
>
> Regarding the naming of each attention mask:
> - **For SG Adaptor:** $\mathbf{M}^{sg}$ represents the attention mask used by **our adaptor**, which leverages scene graph information to refine the embedding of each token. We refer to this as the *Token-Triplet Attention Mask*.
> - **For CLIP Text Encoder:** The other two attention masks, $\mathbf{M}$ and $\mathbf{M}^{\tau}$, are associated with the CLIP text encoder. Specifically:
>   - $\mathbf{M}^\text{causal}$: The default causal attention mask.
>   - $\mathbf{M}^{\tau}$: A modified attention mask informed by scene graph information, referred to as the *Triplet-Triplet Attention Mask*.
>
> ### Question 5
> **Response**:
> Given the role of our adaptor $f(\cdot)$, the attention mask should be $\mathbf{M}^\textbf{sg}$. The previous use of $\mathbf{M}$ was a placeholder. To avoid confusion, we have replaced all instances of $\mathbf{M}$ with $\mathbf{M}^\textbf{sg}$ throughout the manuscript.
>
> ### Question 6
> **Response**:
> Table 2 and Line 413 discuss the ablation study of our proposed SG Mask $\mathbf{M}^{sg}$. The mask $\mathbf{M}^{\tau}$ addresses the limitations of causal masks by rectifying contextualization issues within the CLIP forward process. However, as demonstrated in Section 3.1 and Appendix A.1, directly applying $\mathbf{M}^{\tau}$ in the CLIP forward process does not yield the desired results.
>
> To address this, we designed an adaptor and derived a new attention mask, $\mathbf{M}^{sg}$, based on the same underlying idea. This mask is specifically tailored to rectify the output of the CLIP text encoder, ensuring improved performance and adaptability.
>
> ---
>
> We hope these clarifications and additional experiments address your concerns. Thank you for your thoughtful feedback, which has helped us strengthen our work.

---

> > ### Author Response · Authors · 2024-11-24
> >
> > Dear Reviewer,
> >
> > Thank you again for your valuable feedback and thoughtful comments during the discussion phase. We would like to kindly remind you that the discussion period will conclude on **November 26th**. If you have any additional questions, concerns, or clarifications you would like us to address, we would be more than happy to provide prompt responses. Your insights have been instrumental in shaping the final version of our submission, and we greatly appreciate your time and effort in engaging with our work.
> >
> > Thank you for your attention, and we look forward to hearing from you!

---

> > > ### Comment · Reviewer_EbLH · 2024-11-25
> > >
> > > Thank you for addressing my concerns about the notations.

---

### Author Response · Authors · 2024-12-03
**General Response**

We sincerely thank all reviewers for their valuable comments and insightful feedback. We appreciate that reviewers recognise both the experimental results(reviewer EbLH-"**improvement is significant**", 3Tge-"**outperforms SOTA methods**", Bw61-"**good results**") and motivation(Bw61-"**motivation is straightforward**", EbLH-"**method is straightforward**") of SG-Adapter. We have revised our paper based on the reviewers' comments, with changes highlighted in blue in the updated PDF. Before the detailed response to each reviewer, we kindly request that the reviewers and the Area Chair review this blog content first. It primarily addresses potential misunderstandings and common concerns regarding our paper. For specific questions, we have provided detailed responses in each reviewer's section.

## 1.Citation Format Issue (for reviewer EbLH and 3Tge)
We appreciate reviewer EbLH and 3Tge for their careful review and we have corrected the format of citations in our revised manuscript.

## 2.Experiment Results on Large Scale Dataset(for reviewer 3Tge and wQ1r)
**We have included experiment results on large scale dataset flickr30k[1] in our initial manuscript Section 4.6, Table 3 and Figure 4**. Furthermore, during the rebuttal period, we extend such results by combining 2 more baselines and 3 more metrics to comprehensively demonstrate the generalization performance of SG-Adaptor. We believe that the updated Table 3 as follows could fairly show the effectiveness of our methods:
| **Method** | **SG2IM[2]** | **PasteGAN[3]** | **SGDiff[4]** | **SceneGenie[5]** | **R3CD(AAAI24)[6]** | **ISGC(24)[7]** | **SG-Adapter** |
|------------|--------------|-----------------|---------------|-------------------|---------------------|-----------------|----------------|
| **FID** $\downarrow$ | 99.1 | 79.1 | 36.2 | 62.4 | 32.9 | 38.1 | **25.1** |
| **Inception Score** $\uparrow$ | 8.20 | 12.3 | 17.8 | 21.5 | 19.5 | 30.2 | **57.8** |
| **SG-IoU** $\uparrow$ | 0.085 | 0.091 | 0.122 | - | - | - | **0.413** |
| **Entity-IoU** $\uparrow$ | 0.297 | 0.382 | 0.436 | - | - | - | **0.729** |
| **Relation-IoU** $\uparrow$ | 0.253 | 0.297 | 0.394 | - | - | - | **0.681** |

For R3CD, ISGC and SceneGenie, we just report the inception score and FID from their paper since they do not provide the code.

## 3.Generation of specified application scenarios(for reviewer 3Tge and Bw61)
To validate the generalization performance of our SG-Adapter, **we successfully generate every specified case and scenario as the reviews requests**: case "panda riding on horse" and scenario "crowded scenarios" for reviewer Bw61; scenario "color bind problem" for 3Tge. We have present detailed results in Appendix A.7 Figure 7, Figure 8 and Figure 9 in the revised manuscirpt.


[1] Flickr30k entities: Collecting region-to-phrase correspondences for richer image-to-sentence model.
[2] Image generation from scene graphs.
[3] Pastegan: A semiparametric method to generate image from scene graph.
[4] Diffusion-based scene graph to image generation with masked contrastive pre-training.
[5] Scene graph to image synthesis via knowledge consensus.
[6] R3cd: Scene graph to image generation with relation-aware compositional contrastive control diffusion.
[7] Scene graph to image synthesis: Integrating clip guidance with graph conditioning in diffusion models.

---

### Meta-Review · Area_Chair_S7si · 2024-12-18

**Metareview:**

While both R1 and R2 viewed the paper marginally above the acceptance threshold (based on the simpleness and effectiveness of the method in correcting the incorrect contextualization in text embeddings and enhancing the structural semantics generation capabilities), they shared concerns about poor presentation (i.e., unclear figures, confusing notation, and citation format). R2 also questioned the need for high-quality datasets and asked for more quantitative results (which are addressed by the authors).  R3 and R4 leaned toward rejection of the paper due to its limited novelty, insufficient experimental evaluation, and narrow generalization. With the introduction of SG-Adapter and a new dataset, the method is considered incremental and overly simplistic, failing to provide a fundamental advancement in the field. While the authors responded to both R3 and R4 in addressing the above concerns, both reviewers read the responses and decided marginally below the acceptance threshold. Weighing all the factors and the review quality, I tend to agree with R3 and R4 and rate the paper marginally below the acceptance threshold.

**Additional Comments On Reviewer Discussion:**

See my comments above.

---

### Decision · Program_Chairs · 2025-01-22

Reject